# Influence of Pine and Alder Woodchips Storage Method on the Chemical Composition and Sugar Yield in Liquid Biofuel Production

**DOI:** 10.3390/polym14173495

**Published:** 2022-08-26

**Authors:** Dominika Szadkowska, Radosław Auriga, Anna Lesiak, Jan Szadkowski, Monika Marchwicka

**Affiliations:** Institute of Wood Sciences and Furniture, Warsaw University of Life Sciences—SGGW, ul. Nowoursynowska 159, 02-776 Warsaw, Poland

**Keywords:** alder, biofuels hydrolysis, wood biomass composition, pine, storage method

## Abstract

The aim of this study was to investigate the effect of storing methods of woodchips from two species, pine (*Pinus sylvestris* L.) and alder (*Alnus* Mill.), on the basic chemical composition and sugar yield in liquid biofuel production. Two methods of storing woody biomass were used in the study—an open pile and a cover pile. The wood was felled at the end of November and was stored as industrial chips for eight months from December onward. After this time, material was collected for chemical composition analyses and enzymatic hydrolysis. The results of the chemical composition analysis of the wood for both studied species showed the influence of the type of storage on the composition of the individual structural components of the wood. Based on the results of the enzymatic hydrolysis of the woody biomass, it can be seen that, irrespective of the hydrolysed material (wood, cellulose, holocellulose), the material from the biomass stored in the open pile gave higher results. The hydrolysis efficiency also increased with time, independent of the type of material that was hydrolysed. The highest sugar yield from the enzymatic hydrolysis of wood was obtained for alder wood stored in an open pile. The highest sugar yield from the enzymatic hydrolysis of cellulose was obtained for cellulose extracted from alder wood—as well—that had been stored in an open pile.

## 1. Introduction

Following the confirmation of the effect of global warming in the 1970s, and the subsequent confirmation of the impact of greenhouse gas emissions from industry and transport on this effect, attempts were made to reduce the impact from civilization’s activities on the Earth’s ecosystem through inter-state agreements. As a result, intensive work was undertaken on the development of environmentally-friendly fuels by involving both academics and fuel companies [1,2,3]. Work on fuels has moved in several distinct directions, such as the use of solar energy, wind power, water, geothermal energy, nuclear radiation, and energy from the decomposition of biomass, including plant biomass. The use of plant biomass allows greenhouse gases to be reduced from the atmosphere during plant growth and then it is re-emitted, which in theory creates a closed, environmentally-neutral process [4,5,6].

The efficiency of biofuel production based on plant biomass depends, among other things, on the chemical composition of the material dedicated to its production [7]. This composition mainly depends on the type of biomass (annual plant, perennial plant, grasses including cereals, trees, shrubs, algae, etc.) and the conditions of growth, which consists of the climate, the composition of the soil on which the biomass grows, the water that feeds the plants, the composition of the air from which the plants take up the necessary gases for chemical synthesis, and the compounds that can penetrate and build up in these organisms [1]. The average content of individual chemical compounds is as follows: lignocellulosic biomass of wood origin is cellulose (35–50%), hemicelluloses (20–35%), lignin (15–20%), and ash and others (15–20%) [3,6,8,9]. The technological process itself can also change the compound content of the biomass obtained for biofuel production.

In studies on the acquisition of liquid fuels from lignocellulosic biomass, the following factors are important: the habitat from which the biomass was obtained, the age at which it was obtained, the plant species from which it comes from, and the preliminary treatment to which it was subjected before the enzymatic hydrolysis process (all these elements influence the chemical composition of lignocellulosic biomass and the occurrence of possible inhibitors of the biochemical processes to which it is subjected) [8,9]. This composition varies, of course, depending on the type of plant and its longevity (woody plants have an increase in lignin and mineral content with age, which can be observed from the change in chemical composition at stem height) [10].

Apart from these important elements, the properties of the lignocellulosic biomass that are subjected to hydrolysis and fermentation are influenced by the process and time of storage of the biomass before being processed into fuel. Currently, woody biomass that is used for energy production is co-fired with traditional fuels such as charcoal or lignite. For this purpose, biomass in the form of woodchips is usually stored for about one to two years [11,12,13].

A good example of the storage of woody biomass for fuel purposes is the method of storing biomass used in the wood industry, with a particular focus on paper mills and particleboard factories. Due to the lower amount of change in the raw material during storage, it is assumed that the most favourable storage of wood is in its untrimmed form (wood billets, logs) or sawn as dried lumber [14]. However, due to costs and limited storage space, most of the material in paper mills and particleboard factories is stored in its shredded form (chips or woodchips). This material can be stored in specially prepared warehouses (storage halls, metal silos, or concrete silos) where the material should be dried before storage or prepared in the storage yard where the material does not need to be pre-dried. All forms of storage of shredded wood material require the use of mixing due to the self-steaming process of the material and the decay processes that can occur with long storage. Due to the costs associated with the preparation of storage areas, the most common form of shredded wood storage is in the form of heaps on a concrete storage yard (Figure 1) where the height of the heaps depends on the amount of material and the mixing is carried out by various forms of loaders [11,15,16]. Due to the parameters of the material, we recommend storing shredded wood for up to four months [11]. However, due to the limited availability of material, factories store material from three months up to several years.

Figure 1a,b present images from Google Maps of the storage yards of two wood processing. Figure 1a shows the pulp plant belonging to the Kronospan Sp. z.o.o. group, and Figure 1b shows the particleboard plant belonging to the Pfleiderer S. A. group. These plants belong to multinational wood processing corporations. As can be seen in the figures shown, the storage of woodchips takes place on a paved storage yard.

The disadvantageous decline of shredded woody biomass from the point of view of the wood industry may be beneficial for factories producing liquid biofuels based on lignocellulosic biomass. Additionally, during the storage process, environmentally-friendly pre-treatment is possible [17]. During long-term storage of woodchips in open storage, there is a decrease in extractives, a decrease in hemicelluloses, an equalisation of moisture content, and a change in porous structure [17,18,19]. Changes to the composition and structure of the porous biomass during storage occur spontaneously as a result of biotic and abiotic factors, such as atmospheric precipitation, changes in temperature and humidity (which affect the pores in the biomass, particularly the mesopores), UV radiation and surface ozone (which break down the lignin), and the action of material-degrading enzymes from fungi or bacteria that have infested the material in the landfill [19,20,21,22,23,24,25].

From the point of view of the production process of liquid biofuels, such as bioethanol based on lignocellulosic material such as wood, an important part of the process is the hydrolysis of biomass into simple sugars. This process determines the production of the main products for final processing, i.e., simple sugars, which can then be fermented to produce bioethanol or be chemically synthesized to produce other types of liquid fuels such as furan fuels [26,27].

The use of enzymatic hydrolysis in industry is beneficial for environmental protection. This process is not as efficient and fast as acid or alkaline hydrolysis, while it does not lead to the formation of harmful compounds and waste products that must then be stored or disposed of (as occurs with acid or alkaline hydrolysis). The enzymes used in the process can be returned to the process and also used as a source of high-quality protein in livestock feed [2,26,27,28].

The rate and effect of enzymatic hydrolysis depends on the chemical composition of the lignocellulosic biomass; the availability of polysaccharide chains, such as cellulose and hemicelluloses, for their action; the temperature and pH of the environment in which the process is carried out; and the structure of the material being hydrolysed. In order to improve the properties of biomass, pre-treatments are often used that can improve selected parameters of the material but can adversely affect the content of compounds that are considered to be non-ubiquitous to the biological process. Among the most beneficial pre-treatments are the micronisation process, liquid hot water (LHW), and steam explosion (SE) [2,26,27,28,29]. In the above work, however, no additional pre-treatments were used to reduce the number of factors affecting the enzymatic hydrolysis process.

The aim of the research was to verify the influence of wood storage under selected conditions on the efficiency of enzymatic hydrolysis. Hydrolysis was carried out on three different materials (cellulose, holocellulose, wood) to determine whether the storage has an impact on the process depending on the starting material used. Industrial pine and alder chips were used as wood biomass. This is used as a material by wood processing plants in the production of particleboard.

## 2. Materials and Methods

### 2.1. Wood Biomass

The research was carried out with the use of softwood—pine (*Pinus silvestris* L.)—and hardwood—alder (*Alnus* Mill.). The wood of both species came from the Lubniewice Forest District (Poland). Wood biomass in the form of woodchips was stored on a concrete base for 8 months (from December to July) in an open pile (A) and cover pile (B) (under a metal roof with a pitch of 5°). The temperature difference during storage was 44.3 °C (−9.8 °C in January to 34.5 °C in June). The raw material was stored at the test site in Poland in the Lubuskie Region, Krzeszyce district (52°33′59″ N 15°07′38″ E).

Three samples of woodchips were taken from different locations in each pile then mixed together. For chemical analyses, including the enzymatic hydrolysis process, the particle fraction passing through a sieve with a mesh diameter of 1.02 mm and remaining on a sieve with a mesh diameter of 0.43 mm was obtained.

### 2.2. Methods

The wood particles were extracted in a mixture of chloroform-ethanol (93:7 *v*/*v*) [30]. Extraction was performed on 10-g samples in a Soxhlet apparatus for 12 h. Extracted wood particles were used to determine the cellulose, holocellulose, and lignin content. The cellulose content was determined using the Kürschner–Hoffer method [31]. The holocellulose content was determined by acid chlorite delignification using sodium chlorite in an acid medium [32]. The hemicelluloses content was determined as the difference between the holocellulose and cellulose contents. The acid-insoluble lignin was determined in accordance with TAPPI T222 om-15 [33] and the acid-soluble lignin in accordance with the NREL/TP-510-42618 procedure using a UV/VIS wave length of 205 nm [34]. The summary lignin content is shown in Table 1. The cellulose (1 g) and holocellulose (2 g) samples were used as enzymatic hydrolysis material. The enzymatic hydrolysis was carried out at 45 °C using a buffer of pH 5.4, which consisted of citric acid, sodium hydrogen phosphate, and a mixture of commercial enzymes of Dyadic company—Dyadic Cellulase CP CONC and Dyadic Xylanase 2 XP. The samples were taken after 22, 46, and 70 h. Each determination was repeated three times.

High-performance liquid chromatography (HPLC) was used to analyze the sugars obtained from enzymatic hydrolysis. A universal column Luna NH2 from Phenomenex was used. The analysis conditions were as follows: the flow was 1.5 cm^3^/min, the temperature was set to 50 °C, the eluent was an acetonitrile–water mixture (75:25 *v*/*v*). The hydrolysis yield for each process was calculated separately for glucose and xylose and then summed up and converted into percentages in relation to the dry matter of the individual materials used in the process.

### 2.3. Statistical Analysis

A statistical analysis was carried out on the enzymatic hydrolysis results obtained in order to verify the influence of factors such as species, time, and storage method on the results obtained. For this purpose, an analysis of variance was performed (α = 0.05). The Tukey test was used to compare means, with significance set at *p* < 0.05. Statistica 13 (TIBCO Software Inc., Palo Alto, CA, USA) was used for statistical determinations [35].

## 3. Results

As presented in the introduction, the process of wood biomass storage is associated with a change in the chemical parameters. The chemical composition of woodchips after 8-months of storage in an open pile and cover pile was shown in Table 1. When analyzing the chemical composition, it was noticed that in the case of pine wood, the main difference in the chemical composition between the raw material stored in the open pile and in the cover pile was the lignin content. The pine raw material stored in the cover pile was characterized by a 4.7 percentage point (p.p.) higher lignin content than the raw material stored in an open pile. The cellulose content had a similar level of lignin content for both storage methods of pine raw material, being 47.7% and 47.3% for an open pile and cover pile, respectively. The observed change in the holocellulose content in the pine biomass results from the lower hemicellulose content in the material from the cover pile. Pine biomass from the cover pile was characterized by a higher hemicellulose content, which was 3.6 p.p. more than the stored biomass from the open pile.

From the perspective of high-yield enzymatic hydrolysis, where cellulose is the main source of simple sugars, the method of storage has no significant effect in the case of pine wood. However, a lower lignin content was obtained for material stored in an open pile, which can improve the enzymatic hydrolysis efficiency.

For alder wood, the trend is different. A higher cellulose content was observed when the raw material was stored in a cover pile. The holocellulose content had a similar level for both methods of alder raw material storage, being 70.1% and 71.9% for an open and cover pile, respectively. When the cellulose content changes, the cover pile content is 3.6 pp higher than the open pile, and the lignin (open pile was 37.1% and cover pile was 35.9%) and hemicellulose content was determined by the difference between cellulose and hemicelluloses (open pile was 24.4% and cover pile was 22.1%).

The content of structural components, such as hemicelluloses and lignin, for both types of wood was different from the literature data. When analyzing the results for open-pile pine and cover-pile alder, it can be noticed that there is a decrease in hemicelluloses and lignin content in them compared to the opposite, i.e., reverse storage, respectively. In the case of alder, the decrease in the content of hemicelluloses and lignin in the cover pile leads to an apparent increase in the content of cellulose. This may indicate that the open-pile material and the cover-pile alder have been infected with white decay fungus. These fungi break down lignin and hemicelluloses in order to supply enough carbon for their growth [36,37,38,39,40].

The alder raw material stored in the cover pile and the pine in the open pile were characterized by a lower lignin content, which is, as it was mentioned, beneficial for the efficiency of enzymatic hydrolysis—due to the higher availability of sugars for enzymes [41,42,43].

Figure 2 shows the sugar content obtained after the enzymatic hydrolysis of cellulose, holocellulose, and wood, respectively. Figure 2 shows the efficiency of cellulose hydrolysis with a mixture of commercial enzymes—Dyadic Cellulase CP CONC and Dyadic Xylanase 2 XP, depending on the hydrolysis time (22 h, 46 h, 70 h). The results from obtaining glucose from cellulose for all variants of the analyzed materials were the highest for the hydrolysis carried out for 70 h. This value for pine biomass from an open pile and alder biomass from an open pile and cover pile were over 50%. On the other hand, it was the lowest for pine with a cover pile and amounts to about 43%.

The results of cellulose enzymatic hydrolysis, independent of the material, show an increase with the process time. In addition, compared to the chemical composition of the biomass from which it was obtained (Table 1), a higher increase in the content of the hydrolysis efficiency was observed for the material with a lower content of lignin and hemicelluloses. This may be due to the lower contamination of cellulose by lignin and the greater susceptibility of degraded lignin and hemicelluloses to the mixture used in the Kürschner–Hoffer method [44].

Figure 2b shows the efficiency of holocellulose hydrolysis with a mixture of commercial enzymes from the Dyadic company—Dyadic Cellulase CP CONC and Dyadic Xylanase 2 XP, depending on the hydrolysis time (22 h, 46 h, 70 h). Holocellulose is a material that contains, next to cellulose, hemicelluloses and a less chemically-degraded structure of wood. As in the case of cellulose hydrolysis (Figure 2), extending the process time from 22 h to 70 h increases the efficiency of obtaining glucose from the material. The enzymatic hydrolysis results for holocellulose from pine biomass are about 20 p.p. lower than from alder biomass.

By comparing the efficiency of the enzymatic hydrolysis of holocellulose (Figure 2b) with the chemical composition (Table 1), a higher efficiency of hydrolysis can be observed for the material obtained from pine biomass from the open pile compared to the cover pile. Although the biomass from the cover pile has a higher hemicelluloses content. This may be related to the lower lignin content, which in the case of the pine open pile is about 5 p.p. lower than the cover pile pine biomass.

When comparing the efficiency of holocellulose hydrolysis (Figure 2b) with the chemical composition of wood biomass (Table 1) for alder biomass, the material obtained from the open pile with a higher content of hemicelluloses and lignin gives a higher efficiency of hydrolysis than the material derived from the cover pile, where the difference in efficiency is 4 p.p.

The difference in the hydrolysis of holocellulose from softwood (pine) and hardwood (alder) may be due to the difference in the wood structure and chemical composition of these species. Additionally, coniferous species of conifers are less susceptible to the action of hydrolytic enzymes than deciduous trees due to the specific chemical compounds present in them [45,46].

Figure 2c shows the efficiency of wood hydrolysis with a mixture of commercial enzymes from the Dyadic company, where the enzymes depend on the hydrolysis time (22 h, 46 h, 70 h). As in the case of cellulose hydrolysis (Figure 2a) and holocellulose hydrolysis (Figure 2b), the lowest hydrolysis efficiency for all biomass variants was obtained for 22 h. However, the highest yields were obtained for a hydrolysis time of 70 h. The differences in the hydrolysis of pine wood for 70 h and 22 h were almost 10 p.p. for the open pile and cover pile, and in the case of alder, they were 15 p.p. from an open pile and a cover pile

The highest efficiency of wood hydrolysis was obtained for open-pile alder biomass and was equal to 30.5%. The result for the alder with the cover pile was 3 p.p lower. The highest result for pine biomass was obtained for open-pile biomass, amounting to 22.5%, being higher than the pine material with a cover pile by 3.5 p.p, and lower than the highest (open-pile alder biomass) by 8 p.p.

Comparing the data from Table 1 with the results from the enzymatic hydrolysis of wood (Figure 2c), the results show that the highest efficiency of hydrolysis of wood biomass obtained from the open-pile alder wood is characterized by a higher content of lignin and hemicelluloses than those of cover-pile alder, including having a lower cellulose content. On the other hand, the lowest efficiency of hydrolysis was obtained for pine wood from a cover pile. This biomass is characterized by a higher content of lignin and hemicelluloses than open-pile pine biomass. The cellulose content for both materials is similar. Many factors affect the efficiency of enzymatic hydrolysis. Therefore, it’s hard to determine the influence of single factors when many changes occur, all of which can influence the sugar yield. Many changes occur during storage of woodchips that can influence enzymatic hydrolysis, and in a hydrolyzed system, they can all affect the digestibility of the enzymes. There are lower hydrolysis results for samples obtained from pine than from alder, which may be due to the different composition of monosaccharides in each species. There is more galactane and araban in pine wood than in alder wood. Dyadic enzymes do not contain the alpha-l-arabinofuranoside and do contain small amounts of galactosidase. The reason for this difference may be the inability to spread the sugars that are found in the lignocellulose skeleton [47].

As Tripathi et al. [48] indicate, the storage of wood material to cover the required production in the wood processing sector affects its chemical composition and the parameters of the acquired intermediate product for further processing. Storage of wood in the form of thick chips limits the decrease of stored wood mass [49]. Furthermore, the longer the storage process, the greater the loss of wood biomass [50]. The results of the change in the chemical composition of wood after storage regardless of its method (Table 1) indicate, according to the literature data, an increase in lignin content in the analyzed material in relation to the cited literature data of unstacked wood [25,48]. The change in cellulose content depends, to a large extent, on the species of wood from which it was obtained and not on the method of biomass storage [25]. Hemicelluloses are a structural component susceptible to change in content as a result of storage next to extraction substances [48].

The Tukey test divided the results obtained into seven groups (visible in Figure 2, groups a to g) for the different variants of enzymatic hydrolysis. Most enzymatic hydrolysis results are assigned to the same group for wood hydrolysis for 22 and 46 h. The greatest group variation is observed for the hydrolysis results of holocellulose, indicating the greatest statistically significant variability in the results obtained. The enzymatic hydrolysis of cellulose extracted from stacked woodchips had less variation in the assigned groups, and the results from individual hydrolysis times were assigned more frequently to one group. These conclusions are in line with the observed trend of the obtained yields. They show that there is a statistically significant difference for the individual materials subjected to enzymatic hydrolysis.

The efficiency of enzymatic hydrolysis depends on the availability of polysaccharides for enzyme action, and the peak efficiency of the enzyme (time of enzyme action on biomass) depends on the characteristics of the enzyme. The peak efficiency of commercial enzymes occurs between 46–72 h during the process. After this time, there is a decrease in the efficiency of the process and, in the case of some enzymes, even the disappearance of this process [51,52]. The presented results on Figure 1 show, according to the literature data, that the highest hydrolysis efficiency can be obtained for pure cellulose or cellulose together with hemicellulose (Figure 2a,b). Wood without pre-treatment is a difficult material to hydrolyze, which was also confirmed by other researchers [45,53,54].

An analysis of variance showed that all examined factors (wood species, storage method, and hydrolysis time) had a statistically significant effect on the efficiency of the conducted hydrolysis (*p* < 0.005) (Table 2). In addition, it should be noted that among the studied factors influencing the efficiency of hydrolysis, the time of hydrolysis had the highest percentage coefficient of influence (47.06–68.24%). The lowest percentage coefficient of influence was characterized by factors such as the method of storage of the raw material, which did not exceed 6.66%. However, it should be noted that all the factors studied had a higher percentage coefficient of influence than error, which indicates that the study analyzed the most important factors affecting hydrolysis efficiency. A further analysis also showed that in the case of hydrolysis conducted from holocellulose, the species of wood from which the holocellulose was extracted had a significantly higher percentage impact factor than in the case of hydrolysis conducted from wood or cellulose.

## 4. Conclusions

On the basis of the results of the chemical determinations carried out on the wood biomass of pine and alder wood stored on an open pile and cover pile for 8 months, it can be concluded that:The wood storage method plays a significant role in enzymatic hydrolysis efficiency, which has been statistically confirmed;The pine biomass has a lower yield of enzymatic hydrolysis using Dyadic enzymes than alder biomass;Higher yields of enzymatic hydrolysis were obtained for wood biomass from both species being stored on an open pile;The enzymatic hydrolysis yield of cellulose from wood from both species gave comparable values;The enzymatic hydrolysis yield of holocellulose from pine wood gave lower values than holocellulose obtained from alder biomass.The low enzymatic hydrolysis yield of alder and pine wood results in the need for pre-treatment to enhance the efficiency of the whole bioethanol producing process.

## Figures and Tables

**Figure 1 polymers-14-03495-f001:**
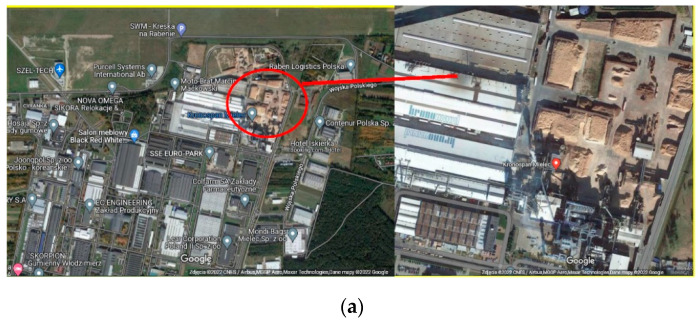
(**a**) Satellite image of the Kronospan Sp. z o.o. group factory in Mielec. (**b**) Satellite image of the Pfleiderer Polska sp. z o.o. factory in Grajewo (www.google.com/maps, accessed on 31 July 2022).

**Figure 2 polymers-14-03495-f002:**
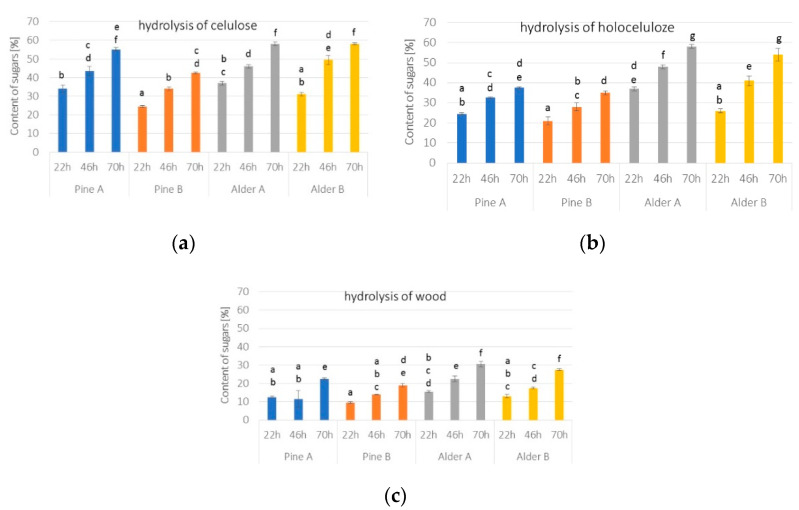
Total sugar yield from enzymatic hydrolysis of: (**a**) Kürschner–Hoffer cel-lulose, (**b**) holocellulose, and (**c**) wood. Letters from a to f in the graph correspond to the groups by Tukey test (α = 0.05).

**Table 1 polymers-14-03495-t001:** Composition of pine and alder wood kept in the open and cover pile.

Component	Pine			Alder		
	Literature Data *	Open Pile (A)	Cover Pile (B)	Literature Data *	Open Pile (A)	Cover Pile (B)
Cellulose (wt% ± SD ***)	44–54	47.7 ± 0.4	47.3 ± 0.3	48	46.2 ± 1.1	49.8 ± 0.1
Holocellulose (wt% ± SD)	63–71	84.8 ± 1,6	88.0 ± 0.5	73–86	70.1 ± 1.8	71.9 ± 1.9
Hemicelluloses **(wt% ± SD)	9–27	37.1 ± 1.3	40.7 ± 0.2	25–38	24.4 ± 2.8	22.1 ± 0.7
Lignin (wt% ± SD))	21–33	34.4 ± 0.9	39.1 ± 0.6	22–24	37.1 ± 0.2	35.9 ± 1.9

* Pattersen, 1984; ** value calculated by the difference in holocellulose and cellulose content, *** standard deviation.

**Table 2 polymers-14-03495-t002:** Analysis of variance of the studied factors affecting hydrolysis efficiency.

Factors	Hydrolysis of					
	Wood		Holocelulose		Celullose	
	p	Pc	p	Pc	p	Pc
Wood spieces (WS)	0.0000	23.48	0.0000	39.62	0.0000	12.83
Type of storage (TS)	0.0001	3.51	0.0000	5.72	0.0000	6.66
Time of hydrolise (TH)	0.0000	63.15	0.0000	47.06	0.0000	68.24
WS × TS	0.0447	0.70	0.0082	0.72	0.0000	5.16
WS × TH	0.0013	2.80	0.0000	3.93	0.0842	0.97
TS × TH	0.2707	0.43	0.0645	0.53	0.1191	0.82
WS × TS × TH	0.0046	2.14	0.1645	0.34	0.0669	1.07
Error		3.77		2.07		4.24

P—probability of error; Pc—percentage of contribution; x—interaction between factors.

## Data Availability

Not applicable.

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
