# Peer review of "Influence of Pine and Alder Woodchips Storage Method on the Chemical Composition and Sugar Yield in Liquid Biofuel Production"

_polymers, 2022, doi:10.3390/polym14173495_

Round 1

Reviewer 1 Report

 In the article titled “Impact of wood biomass storage method on chemical composition and enzymatic hydrolysis yield in bioethanol technology” submitted by Szadkowska et al. they analyzed the changes in wood polymer composition and sugar yield of enzymatic hydrolysis after storing wood biomass.

Although the topic in general is very interesting, the work described in this manuscript can only be attributed as poorly done. The title suggests that the wood biomass was fermented after storage to obtain ethanol, yet nothing like that is mentioned in the article.

The exact period (which month?) in which the wood was stored is not mentioned. Neither are the conditions (temperature, humidity, ...) all these factors do have a significant influence on the degradation of wood.  

How many individual repetitions were done for each type of wood chips and storage method? Was it only one pile each? If so, there can not extracted any reliable data. How often did the authors repeat the analysis of cellulose and lignin content? Please give this number and include the obtained standard deviation for each result.

Carefully check the grammar and spelling throughout the manuscript. Make sure to use “.” as decimal separator.  

Furthermore, the topic of the article does neither fit the scope of the Special Issue neither does it fit into Polymers.

Line 14: don’t use abbreviations in the abstract

Line 38: lignin content in wood biomass is generally higher than the values stated in the manuscript

Line 43: please check the spelling and grammatic of the sentence

Line 63/64: please check this sentence, it does not make sense

Line 70: did you mean mesopores?

Line 86: Material used is not described well. During which month were the experiments conducted? How many repetitions? Amount of material used? Moisture content? Chemical composition of the material? Initial amount of lignin, cellulose etc? Methods applied should not only be cited but also briefly described. Also here is the question how many repetitions were done for each experiment to get reliable data?

Line 143: please include also the values for the starting material and standard deviation.

Line 166: include number of repetitions and error bars representing the standard deviation into the graphs

The article has serious flaws in writing and presenting the data. The methods described do not indicate a sufficient investigation of the topic. No information about data treatment, repetitions and statistical analysis can be found in the manuscript. I therefore can only recommend the rejection of the submitted manuscript.

Author Response

Thank you very much for the comments that the reviewer made on our article. We have tried to address all comments and make changes to the content of the submitted paper.

Reviewer 1: Although the topic in general is very interesting, the work described in this manuscript can only be attributed as poorly done. The title suggests that the wood biomass was fermented after storage to obtain ethanol, yet nothing like that is mentioned in the article.

The subject of the submitted manuscript was to investigate the effect of the process of storing woody biomass for a period of eight months on the chemical composition and the efficiency of enzymatic hydrolysis as the main factors determining the efficiency of the biofuel production process. If the topic may have been misleading we apologise. We propose to change the topic to: „ Influence of pine and alder woodchip storage method on the chemical composition and yield of extracted saccharides in liquid biofuel production”

Reviewer 1: :The exact period (which month?) in which the wood was stored is not mentioned. Neither are the conditions (temperature, humidity, ...) all these factors do have a significant influence on the degradation of wood. 

Information on storage conditions has been added in line 145.

Reviewer 1: How many individual repetitions were done for each type of wood chips and storage method? Was it only one pile each? If so, there can not extracted any reliable data. How often did the authors repeat the analysis of cellulose and lignin content? Please give this number and include the obtained standard deviation for each result.

This information was supplemented in the manuscript.

 Reviewer 1: Carefully check the grammar and spelling throughout the manuscript. Make sure to use “.” as decimal separator.  

We apologise for the error in the separator used. It has been corrected to a full stop.

Reviewer 1: Furthermore, the topic of the article does neither fit the scope of the Special Issue neither does it fit into Polymers.

We understand the reviewer's reservations regarding the issue raised in our article, which is the storage of wood raw material for biofuel purposes. However, we believe that as authors we have chosen the right journal and special issue. Our choice was dictated by 1. the fact that wood is a natural copolymer composed of polymers such as cellulose, hemicelluloses and lignin. 2. biofuels produced from a lignocellulosic material such as wood with some of the solutions including raw material storage will draw on solutions already found in wood industry factories such as particleboard factories, which we as authors may have insufficiently indicated in the first version of the article.

Reviewer 2 Report

The topic is relevant to current context but needs to incorporate all the modifications suggested in the below mentioned form

             Abstract looks to much generic! There need to be statistically show the significance of the current work. What made you to choose this problem? What are all the salient points about this study? What is the limitations of existing materials? Why wood? Why particularly pine wood and alder wood selected for study? What is the benchmark here? What makes them to be compared?

             Title seems to be simple, need to modify to attract more readers

             Abstract should comprise of background, method, results and conclusion.

             Authors quote hydrolysis process except that there are no significant details mentioned in the abstract – to be precise what all tested? With some statistics in broad perspective such as overall enhancement has to be shown in the abstract

             Abstract overscores the content as limitation is 200 words where as currently its 210 words. The entire abstract has to be re-written to make it easily understandable.

             As per journal guidelines 4 to 6 keywords need to be used  

             References should be mentioned as per the journal guidelines

             Authors have to cross verify the quoted references as not more than 2-3 articles are self-cited. Which looks to be bit on higher side (as it’s not a good practise)

             Out of 41 references only two articles are recent hardly there are references of 2021, 2022. There is quite a good amount of work has been carried out on bioethanol with wood as raw material

·                    Introduction should be focusing on existing materials for hydrolysis process. A comparative study with respect to other material has to be listed along with limitations and gaps

             Source of raw material procured along with few snaps would help readers to work and replicate the studies.

             There is no information about number of samples considered for testing?

             The work seems very limited in terms of hydrolysis process of analysis. There need to be inclusion of statistical or analytical works to substantiate the experimentation.

             Reason to store wood chips for 8 months only? Any standard? Proof for this many days of storage? What parameters considered during these many days to store the raw material? In real-time condition is it possible to maintain?

             Country to country, species to species the data will vary. So, when compare to table 1 data, are all the three cases are comparable? If so, what is the common platform to compare?

             What is the significance to keep the wood chips for 8 months? A typical study at each month interval or 15 days interval would have some clarity and possibility to differentiate the entire study process.

             As per table 1, lignin content got enhanced, mechanically when we say lignin increase it will add more strength to the substrate? What parameters are expected out of this study to keep so long..

             Hydrolysis process of hemicellulose and holocellulose seems quite similar is this correct? How to justify the results? Content of sugar will allow the wood to degrade early… will it help for bioethanol process..

             Any characterization study such as FT-IR, EDX, SEM and TEM would have helped in detail to justify the results.

             For every result extracted there need to be standard deviation or tolerance value …

            Any statistical/simulation tool usage to reduce the experimental work or to reduce the number of parameters which are non-critical or significant while testing.

            Best practise while mentioning the graphs to maintain a uniformity of units (SI unit usually preferred)

            Results and discussions are acceptable only with any correlation between the current used material and existing materials in public domain

            There is no comparative study with analytical/experimental/simulation for validation of the extracted results

             The discussion section lags in explanation with respect to the work carried out.  there are limited citations in discussion section to compare the work with existing materials

             The work seems limited and doesn’t have validation or verification with any other statistical/simulation work is missing?

             Conclusion looks to be generic need to compile the outcomes and state based on the tests conducted and convey how best this can fit in the current context for any application.

             In conclusion section, values have to be displayed with explanation. It's better to mention the salient features of the entire work in terms of bullet points with current context

Author Response

Thank you very much for the comments that the reviewer made on our article. We have tried to address all comments and make changes to the content of the submitted paper.

Reviewer 2: ●             Abstract looks to much generic! There need to be statistically show the significance of the current work. What made you to choose this problem? What are all the salient points about this study? What is the limitations of existing materials? Why wood? Why particularly pine wood and alder wood selected for study? What is the benchmark here? What makes them to be compared?

We have tried to restructure the abstract to meet the requirements of the journal and the reviewer's suggestion. The abstract has been completely rebuilt.

Reviewer 2: ●             Title seems to be simple, need to modify to attract more readers

We proposed to change the title

Reviewer 2: ●             Authors have to cross verify the quoted references as not more than 2-3 articles are self-cited. Which looks to be bit on higher side (as it’s not a good practise)

Most auto citations refer to the methodology used for the determinations. Due to the emergence of plagiarism concerning the methodology of determinations carried out in our laboratory

Reviewer 2: ●            Out of 41 references only two articles are recent hardly there are references of 2021, 2022. There is quite a good amount of work has been carried out on bioethanol with wood as raw material

We have added newer literature. However, the topics covered despite being theoretically simple and widely described in the literature are not so. Most of the publications that deal theoretically with wood storage and the impact of this process on the chemical composition deal with the final product, e.g. the amount of energy extracted from energy chips.

Reviewer 2: ●           No information on the number of samples considered for the study? The work seems very limited in terms of the hydrolysis process of the analysis. It is necessary to take into

account the statisticsklIn fact, this was our oversight. We have added the number of repetitions for each operation in the article and enriched the results with statistical analysis.

Reviewer 2: ●             Reason to store wood chips for 8 months only? Any standard? Proof for this many days of storage? What parameters considered during these many days to store the raw material? In real-time condition is it possible to maintain?

We provided information on this in the article. In addition, companies producing ethanol from e.g. maize grain store the maize in micro form in special sleeves for a period of eight months. After this time, they process the dry maize.

Reviewer 2: ●              Country to country, species to species the data will vary. So, when compare to table 1 data, are all the three cases are comparable? If so, what is the common platform to compare?

The chemical composition of wood will always vary depending on the habitat, species and method of determination. For different species and the same determination methods, the composition will be comparable +/- a deviation. Hence, processing plants test the basic parameters on which process performance depends. E.g. a particleboard factory tests the moisture content of the wood and the buffer capacity of the chips (the moisture content influences the drying time, the buffer capacity influences the amount of curing agent used to seal the carpets), an ethanol plant, for example, tests the moisture content and the starch content to calculate how much water to add to the process and how much sugar will be in the mash.

Reviewer 2: ●             As per table 1, lignin content got enhanced, mechanically when we say lignin increase it will add more strength to the substrate? What parameters are expected out of this study to keep so long..

The lignin content of the material increases by decreasing the content of other substances. From the point of view of obtaining bioethanol from wood, this process is disadvantageous because lignin reduces the availability of polysaccharides for enzymes and leads to the formation of compounds considered to be inhibitors. From the point of view of wood-based panel production, an increase in lignin content is beneficial due to the theoretical increase in strength and greater resistance of the material to biological corrosion. However, the degradation of cellulose, which is more responsible for the material's physical properties and mainly its tensile strength, usually occurs during this process.

Reviewer 2: ●             Hydrolysis process of hemicellulose and holocellulose seems quite similar is this correct? How to justify the results? Content of sugar will allow the wood to degrade early… will it help for bioethanol process..

The process of hydrolysis is similar in all cases, while the material undergoing the process differs. Cellulose contains no other compounds, so enzymes have easy access to it and can carry out hydrolysis efficiently. In the case of holocellulose, the material contains cellulose and hemicelluloses, so there are more sugars, but apart from glucose, xylose, arabinose and others are obtained. The material itself is more difficult for enzymes to access, and as a result there can be lower yields of sugars, especially with short process times. The polysaccharides in the wood are the least accessible as, in addition to sugars, lignin and extractives and minerals are present in the material. The lignin and extractives shield the polysaccharides so that the enzymes may have a lower hydrolysis efficiency or have no effect on the biomass.

Reviewer 2: ●             Any characterization study such as FT-IR, EDX, SEM and TEM would have helped in detail to justify the results.

Of course, these results would enrich the work and would be interesting, but at the moment we don’t have enough material to produce such data.

We added the results of the statistical processing to the results obtained. The results are presented for a material commonly used in the particleboard industry as an industry that will compete with wood-based bioethanol plants. In our opinion, processing wood for fuel purposes may have a positive impact on the management of waste from particleboard production, which would increase their competitiveness.

Reviewer 3 Report

Review of the manuscript „Impact of wood biomass storage method on chemical composition and enzymatic hydrolysis yield in bioethanol technology” submitted by Szadkowska et al. to the Polymers journal.

In general the topic is interesting, the number of papers on bioethanol production and the biomass preparation is major and the subject is already well described in the literature, however, in my opinion, there is indeed the lack of knowledge on the influence of the storage method. However, this type of production in my opinion is more associated with the storage in silos than in piles. Anyway, the subject itself seems to be interesting. I was just wondering why the pretreatment was omitted since it is highly beneficial. My suggestions are included below:

Line 1: The form of wood can be specified at this point in the title, wood biomass is a rather broad term used (correctly or not) not only for wood specifically but also for all the forest biomass. You can just say wood chips storage instead wood biomass.

Abstract is informative in general, however, Authors focused on chemical composition of wood and in my opinion the information about enzymatic hydrolysis yield is missing.

Line 14: What is the p.p. (should be explained when used for the first time)?

Line 19-21: this conclusion is strong and doesn’t explained enough. Based on what?

Line 22: Keywords should be listed in the alphabetical order.

In my opinion the Introduction is the weakest part of this paper. First 3 paragraphs need to be rewritten. At this point in my opinion presenting the information such as e.g.: there are research aimed to find an improvement in biofuel technology, the composition of wood is variable, the percentage composition of biomass, all of this is well-known and add only the volume, having no scientific value. I would like to see some specific information about the bioethanol, present some results of the actual studies, not the general sentences. In my opinion presenting this kind of statements is suitable when Authors present some new topic, not the continuation of topic known for years (from another perspective which as I mentioned before I find interesting). If Authors want to present some percentage data maybe it would be beneficial to present it in a tabular way (just a suggestion).

Materials and Methods are mostly clearly presented. The information about the number of replications is missing. The way of calculating hemicelluloses content should be included in methodology, not only the description of table 1. The information about the pile cover is also missing, was this transparent cover or not? Was it roof or just a cover?

Line 95-96: Based on what Authors adjusted the dimensional fraction of powder intended to the analysis?

Results and Discussion is an interesting part of the paper, my suggestions are listed below:

Line 120: What kind of mechanical properties of the biomass are changed?

Line 124-125: What is the possible explanation for changes in lignin content? It seems to be important in terms of the production of bioethanol.

Line 141: Information about the hemicelluloses should be included earlier in methodology. Repeating this sentence is unnecessary.

Table 1: The methodology provides method for determination of both in-soluble lignin and soluble lignin and we have only one parameter in the table.

Table 1: The reference which Authors are using is missing in the bibliography and it makes the “literature data” incomparable. Maybe Authors should look for the newer works assuming the application of the same of similar methods.

What are the reasons for the differences between the species?

Line 153: Have Authors seen the infestation on the surface of the materials? In case of infestation with white-rot or brown-rot fungi during the 8 month storage period the changes would be probably even more visible (acc. to literature)?

The Authors refer to Figures 1 a, 1 b etc. and they are not numbered or signed correctly.

In my opinion the discussion of the results in rather scarce. There are almost no comparison with other studies. The other major issue is a lack of statistical analysis. Taking into account that Authors are writing about the pile of wood chips (each of them probably has a slightly different chemical composition) the statistical analysis in my opinion is necessary.

Author Response

Thank you very much for the comments that the reviewer made on our article. We have tried to address all comments and make changes to the content of the submitted paper.

Reviewer 3: In general the topic is interesting, the number of papers on bioethanol production and the biomass preparation is major and the subject is already well described in the literature, however, in my opinion, there is indeed the lack of knowledge on the influence of the storage method. However, this type of production in my opinion is more associated with the storage in silos than in piles. Anyway, the subject itself seems to be interesting. I was just wondering why the pretreatment was omitted since it is highly beneficial. My suggestions are included below:

The topic of bioethanol production from lignocellulosic biomass is widely researched. However, as a result of learning about the problems of Poland's leading producer of ethanol from maize, we decided to address the impact of storage on the chemical composition and determination of the amount of sugars in the biomass. This producer has different ways of storing maize on its site. The most favourable, from a material point of view, is storage in concrete silos, where mixing takes place automatically by transferring grain between silos using special feeders. Both dry and wet material can be stored in these silos. Metal silos are similar in terms of mixing, but the material can only be stored in dry form as this leads to moulding of the wet material. And storage in special sleeves without mixing the wet material for about 8 months. For financial reasons, building new silos is unfortunately not advantageous. The high construction costs and the metal silos have to be dismantled and erected from scratch every few years. The material is currently stored in sleeves with pre-protection against biological corrosion. In the case of wood processing, companies store woodchips in the manner of open heaps, hence our proposal. We have described it in the paper. The biomass was not pre-treated because we wanted to study how the composition changes during the long storage process itself. Pre-treatment would have introduced further factors that could have affected the yield of the sugars.

Reviewer 3: Line 1: The form of wood can be specified at this point in the title, wood biomass is a rather broad term used (correctly or not) not only for wood specifically but also for all the forest biomass. You can just say wood chips storage instead wood biomass.

We proposed to change the theme of the submitted work to specify the form and type of biomass.

Reviewer 3: Abstract is informative in general, however, Authors focused on chemical composition of wood and in my opinion the information about enzymatic hydrolysis yield is missing. Line 14: What is the p.p. (should be explained when used for the first time)?

Line 19-21: this conclusion is strong and doesn’t explained enough. Based on what?

Line 22: Keywords should be listed in the alphabetical order.

We have improved the volume and content of the abstract as much as possible.

Reviewer 3:

In my opinion the Introduction is the weakest part of this paper. First 3 paragraphs need to be rewritten. At this point in my opinion presenting the information such as e.g.: there are research aimed to find an improvement in biofuel technology, the composition of wood is variable, the percentage composition of biomass, all of this is well-known and add only the volume, having no scientific value. I would like to see some specific information about the bioethanol, present some results of the actual studies, not the general sentences. In my opinion presenting this kind of statements is suitable when Authors present some new topic, not the continuation of topic known for years (from another perspective which as I mentioned before I find interesting). If Authors want to present some percentage data maybe it would be beneficial to present it in a tabular way (just a suggestion).

The introduction has been rebuilt. We have tried to emphasise the issue that prompted us to carry out the research.

Reviewer 3: Materials and Methods are mostly clearly presented. The information about the number of replications is missing. The way of calculating hemicelluloses content should be included in methodology, not only the description of table 1. The information about the pile cover is also missing, was this transparent cover or not? Was it roof or just a cover?

We have tried to fill in the missing information in the methodology.

Reviewer 3: Line 95-96: Based on what Authors adjusted the dimensional fraction of powder intended to the analysis?

The size of the material used in the study was selected based on previous studies and the fraction used in wood chemistry.

Reviewer 3: Line 120: What kind of mechanical properties of the biomass are changed?

A correction has been made to the text. The sentence was general.

Reviewer 3: Line 124-125: What is the possible explanation for changes in lignin content? It seems to be important in terms of the production of bioethanol.

Changes in lignin may be dictated by its decomposition as a result of UV ozone treatment at the surface , and by an apparent increase due to leaching of other chemicals from the wood such as extractives, hemicelluloses. And the effect of fungi, such as moulds, which were not yet visible at the research stage.

Reviewer 3: Line 141: Information about the hemicelluloses should be included earlier in methodology. Repeating this sentence is unnecessary.

It has been corrected in the manuscript.

Reviewer 3: Table 1: The methodology provides method for determination of both in-soluble lignin and soluble lignin and we have only one parameter in the table.

The table shows total lignin, i.e. the sum of soluble and insoluble lignin.

Reviewer 3: Table 1: The reference which Authors are using is missing in the bibliography and it makes the “literature data” incomparable. Maybe Authors should look for the newer works assuming the application of the same of similar methods.

The literature item has been added. We apologise but this error crept in through an oversight between versions of the manuscript.

Reviewer 3: What are the reasons for the differences between the species?

The differences between the species are due to the morphological structure of the two materials studied. Pine is a coniferous species with a lot of resinous substances and its wood is not classified as hardwood. Alder is a deciduous wood with no resinous substances and is considered a hardwood species.

Reviewer 3: Line 153: Have Authors seen the infestation on the surface of the materials? In case of infestation with white-rot or brown-rot fungi during the 8 month storage period the changes would be probably even more visible (acc. to literature)?

We did not observe mycelium characteristic of white or brown decay fungi on the material examined. Under favourable conditions, fungal decomposition of infected material could reach up to 50 % of the material. 

Reviewer 3: The Authors refer to Figures 1 a, 1 b etc. and they are not numbered or signed correctly.

We corrected the figure captions.

Reviewer 3: In my opinion the discussion of the results in rather scarce. There are almost no comparison with other studies. The other major issue is a lack of statistical analysis. Taking into account that Authors are writing about the pile of wood chips (each of them probably has a slightly different chemical composition) the statistical analysis in my opinion is necessary.

In the paper, we have completed the section on the statistical elaboration of the results.

Round 2

Reviewer 1 Report

Thank you very much for implementing the requested information and reworking the article.

The quality has improved significantly and I can now reccomand publishing.

Reviewer 2 Report

The authors have successfully modified the comments except few points left out. I am highlighting these below -

             The discussion section lags in explanation with respect to the work carried out.  there are limited citations in discussion section to compare the work with existing materials

             The work seems limited and doesn’t have validation or verification with any other statistical/simulation work is missing?

Based on the above addressing the manuscript can be considered for acceptance.
